# Update on Anti-Inflammatory Molecular Mechanisms Induced by Oleic Acid

**DOI:** 10.3390/nu15010224

**Published:** 2023-01-01

**Authors:** Consuelo Santa-María, Soledad López-Enríquez, Sergio Montserrat-de la Paz, Isabel Geniz, María Edith Reyes-Quiroz, Manuela Moreno, Francisca Palomares, Francisco Sobrino, Gonzalo Alba

**Affiliations:** 1Departamento de Bioquímica y Biología Molecular, Facultad de Farmacia, Universidad de Seville, 41012 Seville, Spain; 2Departamento de Bioquímica Médica, Biología Molecular e Inmunología, Facultad de Medicina, Universidad de Seville, 41009 Seville, Spain; 3Distrito Sanitario Seville Norte y Aljarafe, Servicio Andaluz de Salud, 41008 Seville, Spain; 4Departamento de Farmacia y Nutrición, Hospital Costa del Sol, 29603 Málaga, Spain

**Keywords:** oleic acid, oleoylethanolamide, macrophages, neutrophils, T cells

## Abstract

In 2010, the Mediterranean diet was recognized by UNESCO as an Intangible Cultural Heritage of Humanity. Olive oil is the most characteristic food of this diet due to its high nutraceutical value. The positive effects of olive oil have often been attributed to its minor components; however, its oleic acid (OA) content (70–80%) is responsible for its many health properties. OA is an effective biomolecule, although the mechanism by which OA mediates beneficial physiological effects is not fully understood. OA influences cell membrane fluidity, receptors, intracellular signaling pathways, and gene expression. OA may directly regulate both the synthesis and activities of antioxidant enzymes. The anti-inflammatory effect may be related to the inhibition of proinflammatory cytokines and the activation of anti-inflammatory ones. The best-characterized mechanism highlights OA as a natural activator of sirtuin 1 (SIRT1). Oleoylethanolamide (OEA), derived from OA, is an endogenous ligand of the peroxisome proliferator-activated receptor alpha (PPARα) nuclear receptor. OEA regulates dietary fat intake and energy homeostasis and has therefore been suggested to be a potential therapeutic agent for the treatment of obesity. OEA has anti-inflammatory and antioxidant effects. The beneficial effects of olive oil may be related to the actions of OEA. New evidence suggests that oleic acid may influence epigenetic mechanisms, opening a new avenue in the exploration of therapies based on these mechanisms. OA can exert beneficial anti-inflammatory effects by regulating microRNA expression. In this review, we examine the cellular reactions and intracellular processes triggered by OA in T cells, macrophages, and neutrophils in order to better understand the immune modulation exerted by OA.

## 1. Introduction

Maintaining an optimal immune system is the best preventive medicine. The major function of the immune system is defense against foreign and/or one’s own malignant cells [1]. Although the immune system depends on genetic factors, other factors may be modulated by lifestyle. These factors include physical exercise, a good emotional state, relationships, meditation, and good nutrition [2], because the immune system is interconnected with the nervous system and the endocrine system [3].

In 2010, the Mediterranean diet was recognized as an Intangible Cultural Heritage of Humanity by UNESCO [4]. In addition, in 2012, it was included by the Food and Agriculture Organization of the United Nations (FAO) in the group of the most sustainable diets in the world [4]. It is considered the most recognized diet for disease prevention and healthy aging, partially due to its demonstrated anti-inflammatory and antioxidative properties, which may impact telomere length [5]. This dietary pattern is characterized by a high consumption of vegetables, olive oil as the main dietary fat, a moderate intake of fish, a low-to-moderate intake of dairy products, a low consumption of meat, and a moderate consumption of wine. Physical exercise is also recommended for the Mediterranean lifestyle [6].

The olive tree (*Olea europaea* L.) is common in the Mediterranean Basin, and it is one of the reasons for the name of this special diet [7]. Olive oil, which is extracted from its fruit, is the most characteristic nutrient in this diet [8]. The beneficial effect of olive oil on health is well-established [9,10,11,12]. The helpful aspects of olive oil have commonly been attributed to its minor components, such as polyphenols, α-tocopherol, and other unsaponifiable compounds [8,13,14], but little attention has been paid to oleic acid (OA) (18:1 n-9 cis-9). This fatty acid is the main component of olive oil (70–80%) [15] and is responsible for many healthy properties [16]. OA is produced via both diet and endogenous synthesis. OA is the most abundant monounsaturated fatty acid (MUFA) in the human diet [17], and, endogenously, it is the main type of monounsaturated omega-9 fatty acid that is formed by stearoyl-CoA desaturase 1 (SCD1), principally from stearic acid (C18:0) by catalyzing Δ^9^ desaturation [18].

OA is the major MUFA in the human circulatory system [17]. In the brain, it is a great component of membrane phospholipids and is highly plentiful in myelin [19]. A significant decrease in OA has been observed in the brains of patients with major depressive disorders and Alzheimer’s disease [20].

OA, like all free fatty acids, has the main function of being an energy molecule and an element of cell membranes. Moreover, since the identification of membrane receptors for free fatty acids (FFAs), new cellular functions have been attributed to it [21]. OA is therefore recognized as a versatile nutraceutical and effective biomolecule. One of its most characteristic effects is its antioxidant capacity because it can directly regulate both the synthesis and activities of antioxidant enzymes [22]. This antioxidant ability may be related to the hypotensive effect attributed to the OA improvement of endothelial dysfunction. Under oxidative stress, the vasodilator molecule oxide nitric is converted to peroxynitrite, producing a hypertensive effect. Another beneficial property is its hypocholesterolemic effect. OA diminishes the expression of cholesterol transport-related proteins, decreases cholesterol absorption [23], and decreases the oxidation of low-density lipoprotein (LDL), preventing atherosclerosis [24].

OA is also recognized as an anticancer molecule because of OA inhibition effects on oncogenes overexpression and its apoptosis effects [25]. OA is generally considered to be an anti-inflammatory molecule, although this quality is debated. Although several authors have reported that OA has an anti-inflammatory effect caused by decreasing well-known oxidative-stress-promoting mediators, for instance, lipopolysaccharides (LPSs), phorbol esters, and cytokines [26,27], others have suggested that it has a proinflammatory effect caused by generating reactive oxygen species (ROS) and by activating the phosphorylation of mitogen-activated protein kinase (MAPK) and/or protein kinase C (PKC) [28,29,30]. The interaction between nutrition and immunology—immunonutrition—is complex because of the fine line between inflammation and anti-inflammation in the maintenance of homeostasis and the prevention of disease [31]. By our experience, we think that OA divergence effects, described in the bibliography, are produced by difference experiment conditions, different cells, concentrations, and time treatment. We results suggest that, OA has a low pro-oxidant effects, but this produces an anti-oxidant response that may be the cause of OA beneficial effects [32].

The aim of this review is to update the knowledge of the molecular mechanisms of OA in the main cells that are part of innate and adaptive immune responses, such as lymphocytes, neutrophils, and macrophages.

## 2. Oleic Acid and Immune Cells

The immune response is composed of a first line of defense, denominate innate immunity, which is characterized by physical and biochemical barriers, together with non-specific cells, such as phagocytic cells (neutrophils and macrophages), dendritic cells, natural killers, and humoral elements. The main mechanism in phagocytic cells is the respiratory burst, which produces ROS to kill microbes in a reaction catalyzed by NADPH oxidase [33]. Furthermore, an adaptive immunity is activated as a second line of defense after the cell-mediated presentation of antigens to B lymphocytes, with the help of T lymphocytes; then, B cells can mediate humoral immunity through the production of high-affinity antibodies and can create immunological memory. Moreover, T lymphocytes can mediate cellular immunity after activation by cytokines or chemokines liberated from helper T cells. The interplay between innate and adaptive immunity is well-recognized [1].

Immunomodulates can induce negative and positive effects. Negative modulation is important in organ transplantations and autoimmune disorders. The positive effect is crucial for restoring and maintaining body homeostasis. Immunomodulatory agents, with antioxidant and anti-inflammatory activities, have attracted great attention as possible preventive agents due to their ability to neutralize chronic inflammation [34]. Long-chain fatty acids (LCFAs) have been implicated in immune modulation [35]. In particular, OA has attracted great attention in recent years as a possible nutraceutical. Preclinical studies have demonstrated the ability of OA to modulate the immune system, affecting both innate and adaptive immunity responses [35]. The effects of OA on signal transduction mechanisms at the plasma membrane, cytoplasm, and nucleus levels are described below.

### 2.1. Oleic Acid and Signal Transduction Mechanisms

#### 2.1.1. Oleic Acid and Cellular Membranes

The first effect of a dietary fatty acid is its incorporation into the lipid bilayer and the changing of its composition, as well as membrane fluidity. This change affects the production of lipid mediators and the interactions of membrane proteins and, thus, influences the signal transduction mechanism [36]. The membranes of immune system cells play important roles in the functions of these cells. In these membranes, the following occurs: the respiratory burst of phagocytic cells, the presentation of antigens in antigen-presenting cells, and the recognition of lymphocyte receptors and all secondary signals exerted by membrane proteins [37].

Olive oil intake increases OA levels in membranes [38], and its twisted chain modifies the interaction within the lipid bilayer, helps to maintain hydration levels, and increases membrane fluidity [39]. Calder et al. described the incorporation of fatty acids into lymphocytes and the effect of fatty acid composition on membrane fluidity. Palmitic and stearic acids decreased fluidity, whereas OA increased fluidity. OA was largely incorporated into phosphatidylcholine [40]. A relatively elevated proportion of OA in membrane phospholipids renders the cell less susceptible to oxidation by decreasing the generation of proinflammatory molecules because arachidonic acid is replaced [41].

In phagocytic cells, membrane fluidity is especially important in determining its phagocytic capacity. Our group described an age-decreasing membrane fluidity in peritoneal rat macrophages and human neutrophils [33,42,43]. Although the OA percentage in these membranes was higher, there are other important fluidity membrane factors, such as increases in the cholesterol/phospholipid ratio and decreases in the proportion of polyunsaturated fatty acids (PUFAs). These factors should have an impact on the final fluidity. We also found an impaired respiratory burst and decreased NADPH oxidase activity with age [44].

Advances have been made in the clarification of the importance of LCFAs in cellular functions with the discovery of their specific membrane receptors (FFA 1–4) [21]. FFA1 and FFA4 are G-protein-coupled receptors for LCFAs, while FFA2 and FFA3 receptors bind to short-chain fatty acids (SCFAs) [21]. FFA1 and FFA4 receptors have been examined in intestinal cells and pancreatic cells due to their significance in obesity and diabetes. Immune cells have been described in the functions of LCFAs and inflammatory processes. Concretely, FFA1 receptor has been found expressed in neutrophils; however, this receptor has not been found in lymphocytes or macrophages. The FFA4 receptor has been found in macrophages and neutrophils but not in lymphocytes [21]. Direct interaction between OA and FFA1 is described in neutrophils. However, the direct interaction has not been described in most immunology cells [21].

#### 2.1.2. Oleic Acid and Cytoplasmatic Signaling Pathways

The effects of OA on signaling pathways and types of immune cells studied are summarized in Table 1. Studies on signal transduction mechanisms have been performed mainly in neutrophils. Hidalgo et al. reported that OA stimulates MAPK phosphorylation, intracellular calcium mobilization, granule release, and superoxide production in bovine neutrophils [30]. Moreover, Carrillo et al. suggested that OA-induced oxidative burst may be a consequence of protein kinase C activation due to an increase in intracellular calcium concentrations in an FFA1-receptor-dependent manner [45]. Thus, Manosalva et al. identified the bovine FFA1 receptor and demonstrated its functional role in neutrophils activated by OA [46]. OA-induced ROS and MMP-9 release are mediated by phospholipase C through both Gq and Gi [30,46,47]. Mena et al. established that this OA-dependent MMP-9 release is also induced by other intracellular signaling pathways, such as p38 MAPK, ERK1/2, and PI3K/Akt [47]. Padovese et al. observed that OA-induced ROS production is determined on NADPH oxidase stimulation, and this increased killing capacity (30%) and phagocytosis (60%) in neutrophils.

In macrophages, OA has an anti-inflammatory action. Karasawa et al. reported that saturated fatty acids induced NLRP3 inflammasome activation in macrophages and induced IL-1β release, whereas OA failed to induce IL-1β release [62]. Oleate protects macrophages from palmitate-induced apoptosis through the downregulation of CD36 expression [51]. Camell and Smith stated that dietary OA increases M2 phenotypic macrophages in the mesenteric adipose tissue of mice [68].

In a previous study, the treatment of J774 cells with non-toxic concentrations of OA had a sustained stimulatory effect on ROS production and increased the fungicidal activity of the cells, suggesting that the enrichment of diets with OA may be beneficial for pathogen elimination. Another study reported that OA in macrophages modulates the post-translational glycosylation of apoprotein E (apoE) in the Golgi apparatus, increasing its secretion [52]. A recent study found that OA decrease, TNFα, IL-6, COX-2, and IL-12 expressions in LPS-stimulated macrophages, showing anti-inflammatory and antifungal properties [53]. Hou et al. reported that OA supplementation increased the AMP/ATP ratio and AMP-activated protein kinase (AMPK) activation and inhibited the NFκB pathway during the inflammatory response to the LPS stimulation of macrophages [54]. These authors suggested that OA might be used for the treatment of sepsis-caused acute liver injury. In the same cells, Hong and Lee found that OA treatments exert anti-inflammatory effects by inhibiting proinflammatory mediators, including PI3K, Akt, MAPKs, NFκB, NOS2, and COX2 [55]. These findings suggest that OA is a potential chemokine-based therapeutic substance for the treatment of the lungs and airway inflammation in allergic disorders. Muller et al. found that OA attenuated LPS-induced prostaglandin E2 (PGE_2_) release. OA significantly diminished the LPS-induced expressions of NOS2, COX2, and IL-6 mRNA. In addition, significant decreases in COX2 and NOS2 protein expressions were also reflected [56].

Recently, Zhang et al. reported that, in phagocytic cells, OA reduced LPS-induced acute kidney injury, improving inflammation and oxidative stress via the Ras/MAPKs/PPARγ signaling pathway [48]. Interestingly, it was demonstrated that OA significantly enhanced the expression of nuclear factor erythroid-2 related factor (Nrf-2), which plays a key role in enhancing cytoprotective genes and antioxidants. Similar to the above, the activities of glutathione peroxidase, superoxide dismutase, and hemooxygenase-1 (HO1) were enhanced in this inflammation disease model after OA treatment. This research demonstrated that OA reduced the expressions of NOS2, COX2, p-p65/p65, and proinflammatory factors (namely, TNF-α, IL-6, and IFN-γ) and elevated the content of IL-10 in the acute kidney injury model. This research found that OA decreased the levels of neutrophils and macrophages in mice with this inflammatory disease [48].

Few studies have been published on OA effects and their signaling pathways in T cells. OA stimulated the proliferation of human lymphocytes isolated from peripheral blood, while other saturated or omega-3 fatty acids decreased it [49]. Similar findings in lymphocytes from tissue adipose have been described [72]. In contrast, Verlengia et al. reported a reduced proliferation of Jurkat T cells treated with this fatty acid [50]. Carrillo et al. showed that OA increased intracellular calcium, a crucial second messenger involved in proliferation and IL-2 expression via the calcineurin/NFAT pathway [58]. The same authors showed that this effect is mediated by an extracellular calcium influx through econazole-insensitive channels [58].

OA is involved in the maintenance of regulatory Treg (T) lymphocyte function. The suppressive function of Treg cells is critical for controlling immune responses and preventing autoimmunity. A recent study found that OA partially restored defects in the suppressive function of Tregs isolated from patients with multiple sclerosis, improving its oxidative phosphorylation metabolism [73].

The best-characterized mechanism highlights OA as a natural activator of Silent Information Regulator 1, sirtuin 1 (SIRT1). This protein is a ubiquitously expressed NAD^+^ deacetylase, with a significant role in preventing inflammation and oxidative stress. Both processes are strongly linked to pathophysiological disorders, such as diabetes, neurodegenerative diseases, and cardiovascular events, and many chronic disorders [74]. SIRT1 is highly expressed in the thymus, supporting the notion that it is associated with immune function regulation [75]. In dendritic cells and macrophages, SIRT1 reduces the formation of inflammatory cytokines [76], and Gao et al. reported that SIRT1 inhibits lung inflammasome activation in a sepsis murine model [66]. The activation of this enzyme has great therapeutic value.

Lim et al. reported that OA stimulates SIRT1 deacetylase activity via the elevation of cAMP intracellular levels and PKA signaling. SIRT1 phosphorylation at Ser-434 elevates its catalytic activity [59]. SIRT1 has numerous targets that may explain its therapeutic potential. SIRT1 deacetylates the NFκB p65 subunit at lysine 310, inhibiting NFκB activity and affecting the nuclear translocation of NFκB and its DNA binding ability [77] (Figure 1). Similar effects have been described for resveratrol, another Mediterranean diet-associated compound and a classic SIRT1 agonist [63].

A direct SIRT1 substrate is the transcriptional coactivator peroxisome proliferator-activated receptor γ coactivator 1-α (PGC1α), which becomes deacetylated and hyperactive after OA treatment but not after treatment with other LCFAs, such as palmitate. This substrate increases genes associated with fatty acid oxidation in an SIRT1-PGC1α-dependent mechanism. OA can therefore be useful in lipid disorders [59].

SIRT1 plays a protective role in Parkinson’s disease [67]. SIRT1 inhibits oxidative stress by maintaining hypoxia inducible factor (HIF-1α) in a deacetylated state. SIRT1 upregulates the expression of forkhead box O3 (FOXO3a) and heat shock factor 1 (HSF-1), inhibiting apoptosis. SIRT1 also reduces the levels of IL-8, IL-6, and TNF-α, inhibiting neuroinflammation. The authors who described the above also investigated the efficacy of a oleic/albumin complex on neuroprotection, suggesting that it is a novel therapeutic molecule that could ameliorate neuronal cell damage in Parkinson’s disease [67].

A novel mechanism proposed for OA is the suppression of the reticulum stress pathway and pyroptosis. OA is able to improve hepatocellular lipotoxicity both in vivo and in vitro via the inhibition of endoplasmic pyroptosis and reticulum stress. Pyroptosis is a new programmed cell death recognized as being caspase-1-dependent and described by plasma membrane rupture and the delivery of proinflammatory intracellular contents, including IL-1β and IL-18. OA substantially alleviated induced endoplasmic reticulum stress and pyroptosis in HepG2 cells [65]. OA alleviated palmitate-induced lipotoxicity in INS-1E cells and enhanced insulin sensitivity in HFD rats. The enrichment of PA-generated ER stress may be responsible for its beneficial consequences in β cells [78].

#### 2.1.3. Oleic Acid and Nuclear Receptors

Peroxisomal proliferator-activated receptors (PPARs) are the main cellular receptors for fatty acids. These nuclear receptors have three isoforms (PPARα, PPARγ, and PPARβ/δ), which are expressed in oxidative tissues to regulate energy homeostasis. In addition, they have also been described in immune cells, playing anti-inflammatory and antiatherogenic roles [79].

The functions of anti-inflammatory PPARs are mediated by several mechanisms, including NFκB inhibition. First, these receptors stimulate the expression of antioxidant enzymes (HO, catalase, and superoxide dismutase); this reduces the intracellular concentration of ROS, which are second messengers in the inflammatory response to activate NFκB. Second, PPARs increase the expressions of IκBα (NFκB inhibitor) and SIRT1. Third, they can directly bind NFκB and induce its proteolytic degradation [80].

Many of the beneficial effects attributed to OA may be exerted via PPAR binding. Not all fatty acids have the same affinity for PPARs. Only fatty acids with 14 or more carbon atoms are able to trigger PPARs [57]. These authors described that, OA has a higher bind affinity with PPARα than PPARγ and δ and higher affinity that other similar fatty acids such as linoic and palmitoleic acid [57]. Additionally, saturated fatty acids (SFAs) with 20 or more carbon atoms do not adjust in the ligand binding domain (LBD) and, consequently, are not activators of PPARs [57]. Double bonds have an essential function in structures with a fatty acid as a ligand. MUFAs in a cis configuration present a better LBD pair than SFAs and fatty acids in a trans configuration of the similar size [80].

Medeiros-De-Moraes et al. found a helpful anti-inflammatory role of OA treatment in sepsis, probably through a PPARγ-related mechanism. OA therapy increased IL-10 concentrations and diminished IL-1*β* and TNF-*α*. Furthermore, neutrophil migration from circulation to the peritoneal cavity and leukocytes rolling on the endothelium were decreased [81].

Although dietary fatty acids do not cross the blood–brain barrier (BBB), it is important to highlight the role of this fatty acid in the brain. Song et al. demonstrated OA neuroprotective effects in rodent models of cerebral ischemia [82]; these neuroprotective effects of OA might be attributable to its anti-inflammatory actions via PPARγ activation [64]. OA is released from brain phospholipids after cerebral ischemia. In the brain, it is a main component of membrane phospholipids and is very concentrated in myelin [19]. OA is a neurotrophic factor, and it stimulates dendrite and axonal development, increases neuronal migration, and promotes synapses. Interestingly, OA levels are decreased in the brains of patients with Alzheimer’s disease and major depressive diseases. Another OA effect is monoamine efflux activation (norepinephrine, dopamine, and serotonin) in the hypothalamus via PPARα [83].

Other nuclear receptors activated via ligands related to lipid metabolism are liver X receptors (LXRs). These receptors, such as those encoding the ATP-binding cassette (ABC) transporters A1 and G1 and SREBP1c, are crucial for cholesterol homeostasis, regulating gene expression. Furthermore, LXR ligands have considerable anti-inflammatory activities, having a vital function in innate immunity [84]. Additionally, LXR ligands decrease atherosclerosis risk by inhibiting inflammatory agents (COX2, IL-6, IL1b, monocyte chemoattractant proteins, and iNOS) in the artery wall [71].

Our group examined the effect of OA in human neutrophils on the mRNA synthesis of both LXRα and ABCA1 (a reverse cholesterol transporter), and, interestingly, this fatty acid augmented the effects of LXRα ligands on ABCA1 and LXRα expressions but inhibited SREBP1c mRNA levels (a transcription factor that regulates the synthesis of triacylglycerides). In our discovery, the main physiological effect was that OA decreased intracellular lipid levels and inflammation markers, such as ERK1/2 and p38 mitogen-activated protein kinase phosphorylation. Additionally, OA decreased the migration of human neutrophils, another marker of the inflammatory state [32].

Another anti-inflammatory mechanism proposed for OA is glucocorticoid receptor mediation. Pegorano et al. demonstrated that OA-containing semisolid dosage forms exhibit anti-inflammatory effects in vivo via glucocorticoid receptors in a UVB-radiation-induced skin inflammation model. OA anti-inflammatory effect is similar to dexamethasone but without adverse effect. Glucocorticoids (GC) are therapeutic agents widely used to treat many pathologies with inflammatory actions. Its effects are mediated by its binding to the glucocorticoid receptor (GR), which regulates the transcription of different genes, controlling changes in the chromatin structure, the transrepression of pro-inflammatory genes, and the transactivation of anti-inflammatory genes. However, the mechanisms that regulate its effects are not sufficiently known, nor is it how it regulates its undesired effects [85]. This natural compound could be a potential option to treat inflammatory skin disorders without undesired effects [86].

## 3. Oleoylethanolamide

Oleoylethanolamide (OEA), a bioactive lipid, is produced postprandially from dietary oleic acid in the small intestine [87]. It may play an important role in food intake regulation through PPAR signaling and vagus nerve stimulation of the appetite center in the brain. It activates the hedonic pathways of dopamine, histamine, and brain oxytocin [88]. OEA can also be formed in mammalian tissues in two enzymatic steps catalyzed by N-acyltransferase and N-acylphosphatidylethanolamine phospholipase D [89]. This bioactive lipid regulates lipid uptake, lipolysis, and beta-oxidation. A deficiency of N-acylphosphatidylethanolamine phospholipase D in adipocytes leads to obesity [90]. Its role in lipid metabolism makes it a potential therapeutic agent for obesity treatment [91].

OEA is a biomolecule with antioxidant and anti-inflammatory properties (Table 2). It can modulate the immune response at two levels. First, it induces IκB and enhances IL-10 expression via PPARα receptor mediation [91]. Second, OEA, interestingly, can modulate the relationship between PPARα and Toll-like receptors (TLRs). TLR pathway activation can diminish PPAR expression [92]. Nevertheless, OEA PPARα mediation reduces the expression of TLR4 [93]. Payahoo et al. conducted a clinical study and found that OEA supplementation decreased inflammation in patients with obesity by decreasing the serum levels of inflammatory molecules, such as TNF-α and IL-6 [94].

The antioxidant properties of OEA are multifaceted, acting as a scavenger for ROS, as well as increasing the activity of antioxidant enzymes [96]. OEA protects plasma lipoproteins against lipid peroxidation and preserves paraoxonase (PON) activity and plasma antioxidant enzymes [98]. Hu et al. found that OEA protects against acute liver injury via Nrf-2/HO1 activation pathways in mice, suggesting that OEA pretreatment significantly reduces hepatic malondialdehyde (MAD) levels and increases superoxide dismutase (SOD) and glutathione peroxidase (GSH-PX) activities [97]. Additionally, OEA reduced the levels of Bax, Bcl-2, and cleaved caspase-3 expressions, suppressing hepatocyte apoptosis. However, OEA reduced the number of activated intrahepatic macrophages and alleviated the mRNA expressions of proinflammatory factors, including IL-6, TNF-α, and MCP1. Furthermore, OEA obviously reduced the expression of IL-1β in the liver and plasma by inhibiting NLRP3 and caspase-1, which indicates that OEA can suppress the NLRP3 inflammasome pathway [97].

This OEA bifunctional property is essential for the treatment of inflammatory diseases with oxidative stress. Preclinical studies have shown that OEA is a potent anti-inflammatory and antioxidant compound that exerts neuroprotective effects in alcohol abuse. OEA is administered intra-parenterally, entering through the BBB and exerting this action rapidly [100]. Treatment with OEA inhibits the alcohol-induced TLR4-mediated proinflammatory cascade, decreasing proinflammatory cytokines and oxidative and nitrosative stress and, finally, preventing neuronal damage in the frontal cortex. OEA decreases NFκB levels, iNOS, and COX-2 expressions, NO accumulation, and lipid peroxidation in the frontal cortex [99]. Similar to that described above for the liver, OEA anti-inflammatory events could be associated with the inhibition of NFκB PPARα mediation [95] (Figure 1). OEA plays analogous neuroprotective roles in numerous models of neurological disorders and brain injuries [101].

OEA is also effective in other inflammatory diseases. It may be helpful in attenuating inflammation and oxidation in patients with coronavirus infection. It has been proposed that the exogenous administration of OEA could be a homeostatic signal to reduce COVID19 infection and improve patients’ inflammatory status [99]. Additionally, OEA therapy with a restricted diet could decrease inflammation in patients with non-alcoholic fatty liver disorder [91]. Furthermore, OEA treatment improves the glycemic index and insulin resistance, and it may be a helpful supplement to control pre-diabetes status [102]. Moreover, OEA use has been indicated to alleviate dysmenorrhea pain in girls by reducing oxidative stress and inflammatory biomarkers [103]. Interestingly, OEA therapy influences gut microbiota composition and the expression of intestinal cytokines in Peyer’s patches [104].

In conclusion, OEA molecules may be a promising therapeutic agent for weight management and obesity treatment, alcoholism, COVID-19, and many other inflammatory disorders.

Recently the anti-inflammatory effects of another OA metabolite, *cis*-7-hexadecenoic acid (16:1n-9). This is synthesized by human phagocytic cells via β-oxidation of oleic acid and its levels are elevated in lipid droplet-laden monocytes, suggesting that it may constitute a biomarker for foamy cell formation [105,106,107].

## 4. Oleic Acid and Epigenetics

Epigenetics is described as heritable variations in DNA and histones without associated modifications in the nucleotide sequence. The central epigenetic mechanisms include DNA methylation, histone alterations, and noncoding RNAs (such as microRNAs (miRNAs)), and disorders of these may be associated with susceptibility to developing a disease [108]. Epigenetic changes are flexible genomic procedures that can potentially be propagated from one generation to another. This is called “transgenerational epigenetic inheritance”, and it may justify how a person’s health and development can be influenced by the experiences of their parents and grandparents [69]. Diet is the most studied environmental factor in epigenetics. Nutriepigenomics is an emergent scientific area that researches the relationships between nutrition and epigenetics [109]. Fatty acids can regulate gene expression by changing epigenetic mechanisms, consequently having positive or negative impacts on metabolic outcomes [70]. However, the mechanisms underlying the effects of diverse types of fatty acids on epigenetic landmarks have still not been completely identified. Various investigations have shown the results of omega-3 and omega-6 PUFAs on DNA methylation [60,110] and butyric acid associated with histone deacetylation [111].

First, to appreciate the roles of fatty acids in epigenetics diseases, Silva-Martínez et al. evaluated the DNA methylation outline particularly induced by arachidonic acid or OA in cultured cells. In THP-1 monocytes treated with either arachidonic acid or OA, DNA hypermethylation or hypomethylation was induced, respectively [112]. The hypomethylation caused by OA improved the inflammation profile. DNA hypermethylation characterizes atherosclerosis in its initial phases and during the progression of stable vascular lesions, and it may be associated with proinflammatory agents, such as arachidonic and palmitic acids [113]. However, hypomethylating agents can slow the progression of vascular lesions [61].

A second epigenetic mechanism is the acetylation of histones. Fatty acids could serve as an alternative source of acetyl-CoA, thereby affecting epigenetic histone marks, such as histone 3 lysine acetylation. Recently, Schuldt et al. demonstrated that OA-related anti-inflammatory effects in fibroblasts are mediated by histone 3 lysine acetylation associated with increased expressions of anti-inflammatory cytokines [114]. These results suggest that OA effects could be exerted by different mechanism by histone acetyltranferase activation and SIRT-1-independence [115].

A third epigenetic mechanism is exerted by microRNAs (miRNAs), which are single-stranded small noncoding RNA molecules of 20 to 24 nucleotides that regulate gene expression mostly at the post-transcriptional level. It has been established that they participate in essential cell processes and regulate almost 30–80% of genes in the genome. miRNAs are differentially expressed in many tissues and are influenced by several external factors, such as diet. These external factors may be used as therapeutic agents against many different diseases as a means to change miRNA expressions [116]. Immune cells express hundreds of miRNAs and have the potential to broadly influence molecular pathways controlling the development and function of immune responses. The deregulation of specific miRNAs leads to various human diseases, including cancer, metabolic disorders, cardiovascular diseases, liver disease, and immune dysfunction [116].

Specific miRNAs, including miR-155 and let-7b, have been linked to inflammatory responses. miRNA-155 is particularly responsive to many inflammatory stimuli, such as TNFα; IL-1β; interferons; pathogen-associated molecular patterns; damage-associated molecular patterns; and TLRs in various cell types, particularly in monocytes/macrophages [117]. miR-155 is rapidly upregulated by NFκB within the first 12 h of the activation of the inflammatory response. In the same way, let-7b, a modulator of cell proliferation and developmental timing, can mediate immune responses and adjust inflammation. Moreover, let-7b may trigger inflammation and immune responses by activating NFκB and IL-6 downregulation [118].

Marques-Rocha et al. studied the expressions of inflammation-related miRNAs in leukocytes from subjects with metabolic syndrome treated for 8 weeks with a Mediterranean diet-based weight loss program. They found that the expression of miR-155-3p was decreased in these cells, whereas let-7b was strongly upregulated because of the dietary treatment. However, these expressions were not correlated with the expressions of the proinflammatory genes in the immune cells. The same group studied the regulatory roles of let-7b and 155-3p in the expressions of inflammation-associated genes in monocytes, macrophages, and LPS-activated macrophages, and they analyzed the potential modulatory roles of different fatty acids, including OA [119]. Let-7b levels were higher in activated macrophages and OA-incubated macrophages. The same results have been described in CACO cells [120].

The miRNA-mediated regulatory mechanisms involved in gene expression control are complex. However, they open a path of knowledge that will allow the use of nutrients in the regulation of metabolism and in the prevention and treatment of numerous diseases.

## 5. Conclusions

We conclude that OA is an immunomodulator with an anti-inflammatory function that, along with an unsaponifiable fraction from olive oil, supports the use of this dietary fat in the Mediterranean diet. Most studies have been conducted on animals; therefore, further research is necessary to confirm the important properties demonstrated by this molecule and its derivate, OEA, in humans.

## Figures and Tables

**Figure 1 nutrients-15-00224-f001:**
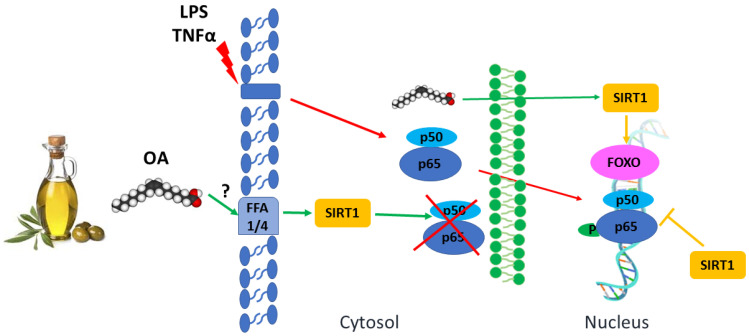
Role of oleic acid (OA) in response to inflammatory stimuli, inhibiting NF-κB signaling pathway by promoting SIRT1 activity on immune cells. Legends: Blunt arrows (┴) indicate inhibition while sharp arrows (→) indicate stimulation.

**Table 1 nutrients-15-00224-t001:** Effects of oleic acid (OA) on signaling pathways, and types of immune cells studied.

General Effects	Specific Effects	Pathways	Cells	References
Pro-inflammatory	↑ ROS	PKC/Ca^+2^	Neutrophils	[43,44]
	↑ Granule release	PKC/Ca^+2^	Neutrophils	[30,45]
	↑ MMP9	MAPK	Neutrophils	[45]
	↑ Phagocytosis	–	Neutrophils	[46]
	↑ Proliferation	Ca^2+^/calcineurin/ NFAT	Lymphocytes	[48,49,50]
Anti-inflammatory	↑ M2	–	Macrophages	[51]
	↓ COX2, TNFα, IL-6, IL-12, NF-κB, iNOS, PGE2	AMPK/MAPK/PI3K	Macrophages/Caco cells/Lung epithelial cells	[52,53,54,55]
	↑ HO-1, GPx, SOD, IL-10 ↓ COX2, TNFα, IL-6, IL-12, NF-κB,	MAPK/Nrf2/PPARγ	Phagocytic cells	[56,57]
	↑ Treg	Oxidative phosphorilation	Lymphocytes	[58]
	↓ Nf-κB	Lys 310 acetilated/SIRT1	Macrophages	[59]
	↑ Let7b	Histone acetilated	Macrophages/Caco cells	[60,61]
Apoptosis	↓ Apoptosis	CD36 expression	Macrophages	[62]
	↑ FOXO3, HSF-1	SIRT1	Neurons	[63]
Neuroprotection	↓ ROS, IL-8, IL-6, TNFα	HIF-1α deacetylate	Neurons (Parkinson’s disease)	[63]
	↑ Monoamino release, dendrites and axon development	PPARγ	Neurons (hypothalamus)	[64]
Lipid metabolism And energy	↓ Lipotoxicity	-----	INS-1 cells	[65]
	↑ Membrane fluidity	Membrane composition	Hep G2 cells	[32]
	↑ AMPK	----	Macrophages	[53]
	↑ β oxidation	PGC1α/SIRT1	Skeletal muscle cells	[66]
	↓ Lipotoxicity	ER stress/pyroptosis/caspase1	Hep G2 cells	[67]
	↑ Apo E secretion	Glycosylation	Macrophages	[68]
	↓ Atherosclerosis lesion	Hipomethylation	THP-1 cells	[69,70]
	↑ LXRα, ABCA1 ↓SREBP1c	MAPK	Neutrophils	[71]
Glycemic Metabolism	↓ IR	ER stress	HFD rats β cells	[65]

**Table 2 nutrients-15-00224-t002:** Effects of oleoylethanolamide (OEA) on signaling pathways, and types of immune cells studied.

General Effects	Specific Effects	Pathways	Cells	Reference
Anti-inflammatory	↑ SOD, GPx	Nrf2/HO-1	Hepatic cells	[92,94]
	↓ Macrophages activation	–	Macrophages	[94]
	↓ IL-6, TNFα, MCP1, IL-1β	NRLP3/caspase 1	Liver/plasma	[93,94]
	↑ IκB, IL-10 ↓ TLR4	PPARα	PBMCs	[89]
Apoptosis	↓ Bax, Bcl2	Caspase 3	Hepatic cells	[94]
Glycemic Metabolism	↓ IR	PPAR	Plasma	[95]
	↓ Food intake	–	–	[89]
Neuroprotection	↓ Alcohol damage	TLR4	Neurons	[96]
	↓ Pro-inflammatory Cytokines/oxidative/nitrosative stress	–	–	[97]
	↓ Neuronal damage, NF-κB, iNOS, COX2, NO, lipid peroxidation	PPAR	Frontal cortex cells	[98,99]

## Data Availability

Not applicable.

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
