# Peer review of "Update on Anti-Inflammatory Molecular Mechanisms Induced by Oleic Acid"

_nutrients, 2023, doi:10.3390/nu15010224_

Round 1
Reviewer 1 Report
This review by Santa-Maria and co-workers summarizes recent advances on the effects of oleic acid on immune cells (OA). The authors describe first a general aspect of OA in immunity, and then specific aspects of the signaling mediated by this fatty acid at the cytoplasm and the nucleus. Possible epigenetic effects are indicated as well. The review is timely, well written and may stimulate research on these emerging concepts.
1) While the review is generally useful and informative, it is also a bit too descriptive and uncritical at times. The authors describe their work and that by others but seldom provide personal views/interpretations or discuss possible implications. For example, lines 80-89, the authors comment on the apparent discrepancies of the effects of OA on a number of innate immune responses mediated by LPS. Although some of the points raised here are further described in the next sections, the authors’ expert opinion on this apparent promiscuity of OA would be appreciated. Likewise, in lines 401-403, the authors mention that the anti-inflammatory effects of OA in fibroblasts are mediated by histone 3 acetylation. How does this relate (if it does) with the OA effects on SIRT1 described in previous sections?
2) The authors include a section to describe the effects of the OA metabolite oleoylethanolamide, which seems appropriate. Recently the anti-inflammatory effects of another OA metabolite, cis-7-hexadecenoic acid (16:1n-9), have been reported (PMID: 29167413; PMID: 27265749). Examination of this literature would also seem appropriate.
3) Minor – line 153, transduction, not transmission.
4) Minor – lines 6-10, Seville, not Sevilla.
Author Response
Point-by-point Reply
Reviewer #1
Comments and Suggestions for Authors
This review by Santa-Maria and co-workers summarizes recent advances on the effects of oleic acid on immune cells (OA). The authors describe first a general aspect of OA in immunity, and then specific aspects of the signaling mediated by this fatty acid at the cytoplasm and the nucleus. Possible epigenetic effects are indicated as well. The review is timely, well written and may stimulate research on these emerging concepts.
1) While the review is generally useful and informative, it is also a bit too descriptive and uncritical at times. The authors describe their work and that by others but seldom provide personal views/interpretations or discuss possible implications. For example, lines 80-89, the authors comment on the apparent discrepancies of the effects of OA on a number of innate immune responses mediated by LPS. Although some of the points raised here are further described in the next sections, the authors' expert opinion on this apparent promiscuity of OA would be appreciated. Likewise, in lines 401-403, the authors mention that the anti-inflammatory effects of OA in fibroblasts are mediated by histone 3 acetylation. How does this relate (if it does) with the OA effects on SIRT1 described in previous sections?
We greatly appreciate this particular reviewer’s comment, and reviewer is right. For this reason, we have revised and rewritten our work including all the appreciations that referee #1 indicated kindly. Moreover, we have discussed the controversial data of the OA immunomodulation in LPS-mediated responses in page 2 lines 90-93.
On the other hand, we have clarify the OA anti-inflammatory effects on fibroblasts page 10 in lines 425-427.
2) The authors include a section to describe the effects of the OA metabolite oleoylethanolamide, which seems appropriate. Recently the anti-inflammatory effects of another OA metabolite, cis-7-hexadecenoic acid (16:1n-9), have been reported (PMID: 29167413; PMID: 27265749). Examination of this literature would also seem appropriate.
We thank reviewer #1, and reviewer is right. For this reason, we have included data reporter by Astudillo A. et al. and Guijas C. et al. Both articles have been revised and included their main results, in page 9-10 lines 391-394
3) Minor – line 153, transduction, not transmission.
We thank reviewer #1 for his/her suggestion to change this word. The change is done in the text.
4) Minor – lines 6-10, Seville, not Sevilla.
We thank reviewer #1 for his/her suggestion to change this word. The change is done in the text.
Reviewer 2 Report
The manuscript reviewed the anti-inflammatory mechanisms of OA and some suggestions are given as followed:
1. Some sentences might lead to different interpretations and are highly recommended to be revised, for example,
Line 79-80, can be described as “due to its effects of oncogenes expression downregulation and apoptosis induction”;
Line 148-149 , “neutrophils have been described in the FFA1 receptor” could be “FFA1 receptor have been found expressed in neutrophils”.
Line 148,“however, they have not been found in lymphocytes or macrophages”, the word “they” stands for FF1A receptors or FFA 1-4?
2. Line 142-150, the delivered information is about FFA, but what’s the relationship between OA and FFA, or is there any evidence directly showed the effects of OA on FFA or direct binding between them.
3. Line 271, is there published data supporting OA binds to PPAR? If yes please cite and describe, if not , as you discussed, not all fatty acids have the same affinity for PPARs, which fatty acids have good affinity to PPARs? And what’s the structure similarity or features enable OA be able/possible to bind to PPARs?
4. Line 309-313, to be more clear and logic to show the anti-inflammatory mechanism of OA through glucocorticoid receptor, how the glucocorticoid receptor regulates the inflammation can be simply described.
5. As described, “Oleoylethanolamide (OEA), a bioactive lipid, is produced postprandially from dietary oleic acid”, and according to the manuscript, OEA can enter through the BBB exerting action easily than OA. Whether the anti-inflammatory function and mechanisms of OA are the results of OEA rather than OA itself, please compare the overlapped function, mechanisms and discuss about these.
6. Line 404-429, all these information is about inflammatory function and mechanisms of miRNA, after this only a few sentences talked about OA without clear relationship or conclusion for OA and miRNA. It is suggested to condense the background information of miRNA and search more literatures to talk more direct or indirect evidences for correlation of OA and miRNA.
7. Figure 1, please increase the font size for mechanism factors.
8. When discussing the mechanisms , what kind of cells or systems were used in the original publications? Are they all in immune cells? This is important because the title highlighted immune cells. As shown in table 2 , not all the studies were in immune cells and as discussed in the manuscript, a lot of information were collected from other cell types or tissue. On the whole, the manuscript was not particularly focusing on immune cells as the title said, it is more like to discuss the anti-inflammatory mechanisms of OA. Therefore, either the title or the information about focused cell types should be revised and improved.
Author Response
Point-by-point Reply
Reviewer #2
The manuscript reviewed the anti-inflammatory mechanisms of OA and some suggestions are given as followed:
- Some sentences might lead to different interpretations and are highly recommended to be revised, for example,
We greatly appreciate this particular reviewer’s comment, and reviewer is right. For this reason, we have revised all manuscript, pain special attention on his/her suggestions.
Line 79-80, can be described as "due to its effects of oncogenes expression downregulation and apoptosis induction";
We thank reviewer #2 for his/her suggestion restructure and clarify this sentence, page 2, lines 79-80
Line 148-149 , "neutrophils have been described in the FFA1 receptor" could be "FFA1 receptor have been found expressed in neutrophils".
We appreciate this particular reviewer’s comment, and reviewer is right. For this reason, we have modify this sentence like reviewer #2 proposes, page 4 line 153
Line 148,"however, they have not been found in lymphocytes or macrophages", the word "they" stands for FF1A receptors or FFA 1-4?
We thank reviewer #2 for his/her the depth analysis in the manuscript, and we have change this particular expression mistake, page 4 line 154.
- Line 142-150, the delivered information is about FFA, but what's the relationship between OA and FFA, or is there any evidence directly showed the effects of OA on FFA or direct binding between them.
We greatly appreciate this particular reviewer’s comment, and reviewer is right. For this reason, we have revised and rewritten our work including all the appreciations that referee #2 indicated kind, page 4 lines 155-157
- Line 271, is there published data supporting OA binds to PPAR? If yes please cite and describe, if not , as you discussed, not all fatty acids have the same affinity for PPARs, which fatty acids have good affinity to PPARs? And what's the structure similarity or features enable OA be able/possible to bind to PPARs?
We thank this reviewer’s comment giving the opportunity to further clarify and improve the manuscript, for this, now described these data in the text, page 7 line 282-284.
- Line 309-313, to be more clear and logic to show the anti-inflammatory mechanism of OA through glucocorticoid receptor, how the glucocorticoid receptor regulates the inflammation can be simply described.
We thank this reviewer’s suggestion and we have change this paragraph and delved more into the regulation of inflammation by these receptor, page 8 lines 323-331.
- As described, "Oleoylethanolamide (OEA), a bioactive lipid, is produced postprandially from dietary oleic acid", and according to the manuscript, OEA can enter through the BBB exerting action easily than OA. Whether the anti-inflammatory function and mechanisms of OA are the results of OEA rather than OA itself, please compare the overlapped function, mechanisms and discuss about these.
We thank this reviewer’s suggestion and we fully agree with him but it is difficult to determine if all the effects of OA are produced by its transformation to OEA, nor the percentage of transformation of this, so we limit ourselves to describing the effects described by the different authors
- Line 404-429, all these information is about inflammatory function and mechanisms of miRNA, after this only a few sentences talked about OA without clear relationship or conclusion for OA and miRNA. It is suggested to condense the background information of miRNA and search more literatures to talk more direct or indirect evidences for correlation of OA and miRNA.
We greatly appreciate this particular reviewer’s comment, but currently there are very few results on the relationship between miRNAs, inflammation and OA. For this reason, in our review we have introduced only the data that seems most relevant and best demonstrated, in our opinion.
- Figure 1, please increase the font size for mechanism factors.
We appreciate this particular reviewer’s comment and we have changed the figure
- When discussing the mechanisms , what kind of cells or systems were used in the original publications? Are they all in immune cells? This is important because the title highlighted immune cells. As shown in table 2 , not all the studies were in immune cells and as discussed in the manuscript, a lot of information were collected from other cell types or tissue. On the whole, the manuscript was not particularly focusing on immune cells as the title said, it is more like to discuss the anti-inflammatory mechanisms of OA. Therefore, either the title or the information about focused cell types should be revised and improved.
We greatly appreciate this particular reviewer’s comment, and we have changed the title.
